# Factors Affecting the Sustained Market Position of TV Channels: Insights from Audience Preferences

**Asım Dinler** [1,*]**, Tarık Atan** [1,*] **and Aysen Berberoglu** [2]

1    Department of Business Administration, Faculty of Economics & Administrative Sciences, Cyprus International University, 99258 Lefkoşa, North Cyprus, Turkey

2    Faculty of Business Administration, University of Mediterranean Karpasia, 99010 Lefkoşa, North Cyprus, Turkey

\*    Correspondence: dinlerasim@gmail.com (A.D.); tatan@ciu.edu.tr (T.A.)

**Abstract:** Presently, in a rapidly changing environment, television (TV) channels—such as other profit-seeking institutions—should strive to gain a competitive advantage over their competitors to save their market shares and sustain themselves in the sector. This especially applies to local TV channels, which not only compete with other local TV channels, but also compete with international and online platforms at the same time. In North Cyprus, where the TV sector is very small and limited, TV channel owners try to generate profit out of a very small market. In this situation, it is important for them to create a good brand image in the minds of the audience, as well as good program quality, their end goal being to sustainably gain the preference of the audience. It is also important for local TV channels to increase their viewing times, leading to the sustainable behavior of the viewers. The present study aims to evaluate the hypothesized relationship between TV channels' image and audience preferences, taking into account the mediation effect of program quality and the moderator effect of viewing frequencies. The scope of the study encompassed people living in North Cyprus who watch local TV channels, from whom a sample of 384 respondents was drawn. An online self-administered questionnaire was distributed in order to collect the data. The collected data were analyzed with the help of SPSS and AMOS 24. The results from the analysis showed that there was a positive relationship between the channel image and channel preferences of the audience, and that this relationship was mediated by TV program quality. On the other hand, viewing frequency was found to have no statistically significant moderation effect on the relationship between channel image and audience preferences.

**Keywords:** TV channel image; program quality; audience preference; sustainability; competitive advantage





## 1. Introduction

Currently, traditional performance indicators—that is, indicators that show the profit levels of companies, such as knowledge ownership and production, the use of technology, the profile and satisfaction of customers, the quality of the product or service produced, assurance, customer loyalty, and the contribution to the environment—have been replaced by criteria that are not concrete, intangible, and difficult to measure [1]. It is now more important to assess how consumers perceive a product or service, with their perceptions deemed to determine the success of companies. A customer's perception, whether favorable or negative, about an object, institution, a company, or even a person, is referred to as an "image", relative to each individual. Sustaining a positive image is crucial; if the image formed in the mind of individuals is negative, then strategies for improving the image to make it positive must be developed [2]. As mentioned above, an image is created in the mind of the individual; therefore, it is an abstract element, and the outcome of its assessment by each individual comes from the individual's own evaluation of the distinction between positive and negative scenarios [3]. When it comes to the TV company sector, the situation is no different; the image of the TV channels, which is created in the minds of individuals

in the audience (who can be considered customers) is important, because there is no doubt that it shapes their attitudes and behaviors towards the TV channels. It is possible to say that the TV channel image is not any different from the brand image of companies. From the marketing perspective, it is possible to accept each TV channel as a brand, and the TV audience can be accepted as the customer. Furthermore, TV programs produced and broadcast by TV channels are products of these brands, and are delivered to the end consumer: the audience. In today's world of competitiveness, increased competition among brands has led to consumer behavior having a significant effect on the sustainability of companies in every sector. TV channels, such as all other companies, should always be ready to acquire a competitive advantage over their competitors in order to remain in the market, achieve sustainable economic growth, and retain their market share. While demand has always been important in shaping the behavior of companies, customers now have more power to influence those companies, markets are now constructed in accordance with customer choices, and brands carry out services in accordance with the behaviors and desires of customers as a result of the increased competition. In this sense, TV channels should invest in and improve their program quality in accordance with audience preferences in order to engage with the audience, increase viewership, and create loyalty. Loyalty is very important, because it leads to repetitive customer behavior. For TV channels, the sustainable behavior of viewers ensures a long-term commitment in terms of behavior and attitude; therefore, repetition and the long-term brand preference of the audience increase the possibility of the development of brand loyalty [4]. There is no doubt that loyalty is the key to companies sustaining their position in the market. Such as other companies, TV channels should supply their audience with the same or a greater value than their competitors, be more efficient in their production (by cutting costs), or elaborate specific activities that generate the greatest final value (such as variety and quality programs) and differentiation [5]. Customer preference for a brand is one of the behavioral dimensions of consumer relations to brands, and the level of a consumer's enthusiasm, loyalty, and inclination towards a brand, when compared to other brands, can be defined as brand preference [6]. In this sense, increased consumer preference is a competitive advantage for TV channels, as it offers the promise of retaining their position in the market. An increase in the positive, subjective, and perceptual associations and phenomena created in the mind of the consumer—who creates the image—can contribute to the consumer's brand preference [4]. Brand preference can be attributed to repeating purchases or usage behaviors, which also contribute to the sustainability of the brands in their sector. Furthermore, TV channels are expected to generate program diversity (product diversity) in response to customer preferences and to adapt in response to the changing environment in order to sustain their position in the market and expand their share. Brands that do not consider the consumer's behavior and preferences are excluded from the market, as they cannot create brand preference and they lose their ability to retain customers [7]. From a marketing perspective, TV channel performance, in terms of program quality and audience satisfaction, leads to a competitive advantage [8].

On the other hand, a brand's image is deemed to affect how a customer perceives the quality of the product. The effect of brand image on perceived quality has been investigated in the extant literature, and it has been determined that brand image has a positive impact on perceived product or service quality [9–12]. It is possible to interpret this finding for TV channels as follows: if a TV channel can positively influence the image the audience has of it, the perceived program quality of the audience concerning the channel's programming would be high [4]. In the TV sector, audiences can have some prejudices and expectations about TV channels and their products (programs) just by looking at the brand name of the TV channel, such as in other sectors. A company's brand image is a key aspect in influencing customers' thoughts, perceptions, and ideas about the brand's products. The image formed in the minds of the audience is a result of the accumulation of beliefs, ideas, and impressions [13]. The situation is similar in the TV channel industry. When a new program is carried on a well-known channel with a good brand image in the minds of the

audience, this new program is already assumed to be of high-quality because the audience is familiar with the quality of the TV channel due to its image.

Furthermore, quality perception is expected to contribute to brand preference and, therefore, positively contribute to the success of companies and help them sustain themselves in the market. Additionally, a well-defined diversification strategy for the programs of TV channels could assist them in creating or sustaining their competitive advantage [14]. Lastly, according to the extant literature, the quality perceived by consumers plays a mediatory role in the relationship between brand image and purchase intention as well [15].

### 1.1. Sustainability in the TV Sector

The present study tries to understand the reasons behind an audience's choices for local TV channels, which is the most important aspect for TV companies to remain competitive in their sector. Local TV channels in North Cyprus try to gain the preference of the audience so they can compete with each other to retain their position in the market. The sustainability term mentioned in the current study is based on the definition of "the quality of being able to continue over a period of time" (Cambridge dictionary). As mentioned before, having continued (repeating) customer preference, in other words, gaining the loyalty of the audience, is the key for companies to sustain their position in the market. Furthermore, as the other subject of the present study indicates, the perception of the quality of TV programs is expected to affect the preferences of the audience, which can have a positive influence on the success of TV companies and help them sustain their market positions. Understanding the nature of these relationships is very important for both managers and owners of TV channels, such as it is for others in the business world, so that they can make plans to shape their investment decisions to have a better channel image to sustain their position in a small market sector such as that in North Cyprus. Additionally, in order to compete in the market, same as in other sectors, TV channels should continuously improve their program quality and diversity to meet audience demand (as mentioned—demand is the key), so they can adapt to the changing environment in order to sustain their position in the market.

### 1.2. Research Gap

When previous studies in the literature were examined, no empirical studies were found in this field regarding the relationships between TV channel image, program quality, viewing frequencies, and audience preference, or how these relationships are related to the sustainability of TV channels in the sector all together. There was only one study found, which was carried out in Turkey regarding its national TV channels [4]. The study looked at the relationships between TV channel image, program quality, and channel preference. Other than this specific study, there were some studies about the viewing rates of channels; more specifically, the program rating scores. In this sense, the present study tried to test completely new hypothesized relationships based on a completely new market. The present study was carried out in North Cyprus, where local TV channel companies are very small and lack investment. This market is very different compared to the market where the only previous study was carried out, because the previous study in the literature was carried out in Turkey, where the population is almost 80 million, which is much higher than that of North Cyprus, with only a population of approximately 350,000 people. In Turkey, there are more than a hundred local TV channels and more than twenty national channels. On the other hand, in North Cyprus, the total number of local TV channels is not even as high as 20. In North Cyprus, there is local competition in the sector, and sustaining growth in the sector is so difficult that companies only aim to keep their market share just to continue to generate profit. Especially now, online streaming and highly competitive TV channels from Turkey are posing great risk for local TV channels in North Cyprus. To gain a competitive advantage over others, TV channel companies in North Cyprus have no choice but to gain a bigger audience by projecting a positive image and convincing them to choose their channel.

### 1.3. Aim of the Study

The aim of the present study is to evaluate the hypothesized relationship between local TV channels' image and audience preferences for the sustainability of TV companies, together with the expected mediation effect of program quality and moderation effect of viewing frequencies. The audience living in North Cyprus and watching local TV channels comprised the universe of the study, and among the whole population, a sample of 384 respondents was selected. An online, self-administered questionnaire was utilized in order to collect the data, which were analyzed with the help of SPSS and AMOS. According to the results, there is a positive relationship between channel image and channel preference among the audience. Additionally, this relationship is mediated by the TV program's quality, and the viewing frequency of the audience was found to have no statistically significant moderation effect on the relationship between channel image and audience preference.

## 2. Literature Review and Hypothesis Development

### 2.1. Defining Variables

In the TV sector, the criteria for quality and how to measure it are very critical hard to define. There is no doubt that quality is one of the most important factors for success, such as in other sectors. The quality of a product or service can be defined as the difference between a customer's expectations and their actual perception [16]. Additionally, how well the product or service matches the expectations of the customer can define its quality [17]. Because quality is a matter of perception, it is solely determined by the customer's judgment [18]. Handoko [19] stated that product quality is the set of features of a product or service that depends on its ability to meet actual or implied customer needs.

The current study emphasized the importance of content (program) quality. The product of local TV channels is only the content they create; in other words, their program, because unlike paid TV channels, it is not possible to consider the service quality, price, resolution, transmission stability, or streaming quality when measuring the quality of unpaid local (satellite) TV channels. Therefore, in the current study, the respondents were only expected to evaluate the program quality of TV channels based on their own perceptions created by their experiences and mostly based on the comparisons they determined with other local TV channels.

On the other hand, another concept of the current study was the image of the TV channels in the minds of customers, or, in other words, the audience. The image considered in this study was the brand image, which is a reflection of how an audience perceives TV channels and what they think and feel about them [20]. Additionally, it was possible to define the brand image as a certain characteristic of a brand that remained in the minds of customers [21].

The brand image is often relied upon by the customers to determine or recognize the quality of the product or service [22]. Respondents in the current study were expected to rate TV channels based on their own emotions.

Lastly, customer preference is another variable in the study. The concept of "preference" itself stands for the attitude of someone towards a choice, which is reflected during a process of decision-making [23]. Moreover, customer preference is the tendency of customers to choose a brand among other valued options and their willingness to continue with the same choice [24].

### 2.2. Relationship between Channel Image and Program Quality

The image is generated by the accumulation of beliefs, ideas, and impressions about the person, thing, or company [13]. It is also possible to say that the image conveys a cognitive meaning that encompasses people's experiences. A cognitive and symbolic assessment is at the forefront of people's (viewers') decision-making process when choosing a product, namely, programs for TV channels [25]. TV channels that can create a positive image in the audience's minds would have a high perceived program quality, which would then in the end contribute to channel preference [4]. It has already been confirmed that

brand image affects how the quality of a product is perceived [15], and the perceived quality tends to affect consumers' behaviors [26].

On the other hand, product quality is determined by the viewer rather than brand managers or quality control specialists. As a result, brands must accurately determine consumer preferences and meet the needs of consumers [27]. It is important for TV channels to create audience value that is different from that of their competitors with the help of quality programs to gain a competitive advantage; gaining a competitive advantage in this manner also leads to better business performance [28].

Creating a good brand image is easier when the perceived quality is high, because the consumer's sense of quality contributes to the brand image [29]. Therefore, if the audience is convinced that a TV channel has high-quality programs, this contributes to a positive brand image in the minds of the audience, and they are more likely to choose this TV channel again. The image also influences how the audience (consumers) perceives the quality of the product (TV program) [15].

**Hypothesis 1.** *There is a relationship between channel image and program quality.*

*2.3. Relationship between Program Quality and Channel Preference*

One of the most essential methods for businesses to reach customers is through perceived quality, and now, brands must provide high-quality goods and services in order to keep up with the times and to gain a competitive advantage in their sector [30]. Previous studies pointed out that perceived quality has an influence on consumers' intentions [26]. In the case of TV channels, an increase in all positive perceptions created in the mind of the viewers about program quality, which also contribute to the image, is expected to increase the consumer's brand preference [4], in other words channel preference for the TV channels.

As a result, when audiences perceive high quality in TV programs, they tend to watch the channel more frequently, contributing to the TV channel's long-term viability in the industry.

**Hypothesis 2.** *There is a relationship between program quality and channel preference.*

*2.4. Relationship between Channel Image and Channel Preference for Sustainability*

TV channels have a responsibility to identify consumer preferences, recognize the changing consumer culture, and meet consumer expectations, desires, and even dreams to boost their channel image if they want to remain in the sector. Furthermore, the TV channel that best meets consumer needs and expectations maintains or even increases their viewing share [7], which is one of the most important factors affecting the competitive advantage of TV channels.

In marketing terminology, TV networks are brands, and viewers are consumers. Programs that are shown on television networks are products. Brand and product competition is on the rise at the moment. In this approach, TV channels are also assessed in terms of brand image, brand loyalty, brand preference, perceived product quality, brand trust, brand value, etc. [4]. Brand image now plays an active role in the choice process, together with the influence of demographic, social, and psychological aspects that consumers have. Each of these aspects has a unique impact and contributes to the formation of brand preference [31] and the sustainability of the brand in the sector among the competitors. Overall, an increase in all subjective perceptions created in the mind of the consumer leads to a positive brand preference among consumers [4]. Therefore, it is possible to say that brand image has a positive and significant impact on consumers' intentions [15], which can lead to a competitive advantage.

**Hypothesis 3.** *There is a relationship between channel image and channel preference.*

*2.5. Relationship between Channel Image and Channel Preference for Sustainability Is Moderated by Daily Viewing Frequencies*

The impression generated in the minds of the target community is referred to as the image, and a favorable company image can enhance the service quality [32]. Brand image influences the intentions of consumers [33], and it is possible to state that brand image has a positive and significant influence on brand preference; Ref. [34] the viewing frequencies of the audience, on the other hand, are expected to have a moderate effect on the audience's preferences. Because a loyal audience is believed to contribute to the longevity of companies, repeat viewing behaviors are anticipated to foster loyalty, which would result in a continuous brand preference [12].

**Hypothesis 4.** *The relationship between channel image and channel preference for sustainability is moderated by daily viewing frequencies.*

As a result, from the review of the literature, the conceptual model shown below in Figure 1, was constructed with four variables.

- **Hypothesis 1:** There is a relationship between channel image and program quality.
- **Hypothesis 2:** There is a relationship between program quality and channel preference.
- **Hypothesis 3:** There is a relationship between channel image and channel preference.
- **Hypothesis 4:** The relationship between channel image and channel preference is moderated by daily viewing frequencies.

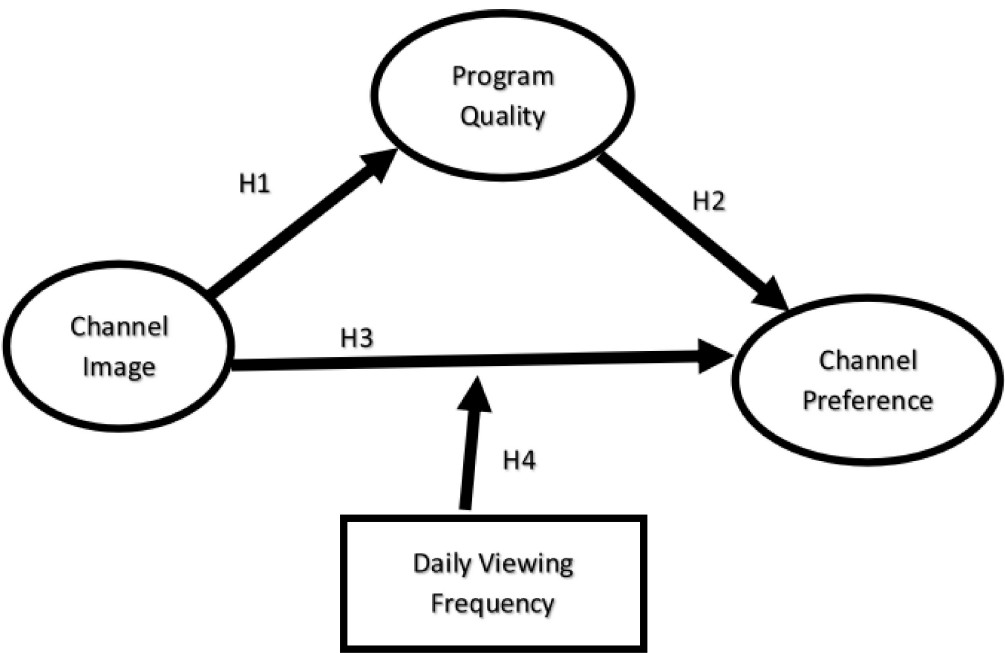

**Figure 1.** Conceptual model.

## 3. Research Methods

*3.1. Population and Sample Size*

The study adopted a quantitative research method. Respondents were introduced to a self-administered questionnaire. Due to the COVID-19 pandemic restrictions, questionnaires were mostly distributed online, but some were filled out on paper. The population of the study comprised above 18-year-old individuals currently residing in North Cyprus who declared to actively watch local TV channels on a daily basis. The sample size was 384 respondents, which was a sufficient number of respondents with a 95% confidence interval and 5 margins of error. The study adopted a convenience sampling method to attempt to sample the entire population.

According to Krejcie and Morgan, 1970 [35], with a 95% confidence level and heterogeneous consideration of the universe, the desired sample size for a universe including 1 million people was calculated as 384. The sample size for our study was also 384, for a population of approximately 350,000.

### 3.2. Measurements

At the stage of forming the survey questions, the author benefited from the studies existing in the literature, which had previously been tested for reliability and validity. However, in order to verify the instrument's validity, a factor analysis was also performed. There were three sections to the created questionnaire.

In the first part of the questionnaire, questions about demographic characteristics were asked of the respondents. This group of questions included gender, age, education level, occupation, and income level.

Later, three questions about the TV viewing habits of the respondents were asked.

In this group of questions, respondents were asked about how many hours of TV they watched on average, which local channels they watched the most, and the reason why they preferred the local TV channel they followed the most.

In the third part of the questionnaire, there were statements regarding channel image, perceived program quality, and channel preference. Respondents were expected to choose one option from the five-point Likert scale.

Statements about channel image perception included the first 9 statements (question 1–9). The next 7 statements were used for the measurement of program quality (questions 10–16). Finally, respondents were asked 3 statements about audience preference (questions 17–19). The 5-point Likert scale system was used to collect the responses, which were listed as "Totally disagree", "Disagree", "I have no idea", "Agree", and "Totally Agree".

The statements used in the questionnaire form were adopted from the work of Gülhayat Şimşek and Fatma Noyan titled "Can Television Preference Be Modeled?" [4].

### 3.3. Research Tools

A questionnaire was used as the survey method, and the questionnaire scale used was summarized (Table 1).

**Table 1.** Variables and number of questions.

| Variable | No. of Questions | Resource |
|---|---|---|
| Channel Image | 9 | [4] |
| Program Quality | 7 | [4] |
| Channel Preference | 3 | [4] |

### 3.4. Analysis Procedure

The collected data were analyzed using SPSS 22 and AMOS 21. SPSS 22 was used for the frequency analysis to analyze the demographic characteristics and create the exploratory factor analysis and reliability analysis for the validity of the latent variables to test the validity between factors for the constructs used for the questionnaire. After removing some variables from the set of questions as a result of the factor analysis, AMOS 21 was used to check the parameter estimates and the GOF (goodness of fit) of the measurement model based on the values of the CMIN/DF (chi-Square value and degree of freedom), GFI (goodness of fit index), AGFI (adjusted goodness of fit index), CFI (comparative fit index), RMSEA (root mean square error of approximation), and IFI (incremental fit index).

Lastly, the hypotheses were tested in order to be accepted or rejected by looking at the causal relationships between the variables for the structural equation modeling analysis.

### 3.5. Demographic Variables and Frequencies

According to the demographic data as shown in Table 2, it was verified that the majority of the respondents were male (52.9%), and the rate of female respondents was 46.6%. Two respondents did not answer the gender question, so 0.5% of the gender data was missing.

**Table 2.** Demographic characteristics of respondents.

| Variable | Percentage (%) | Number (N) |
|---|---|---|
| Gender | | |
| Female | 46.6 | 179 |
| Male | 52.9 | 203 |
| Missing | 0.5 | 2 |
| **Age** | | |
| 17–24 | 38.3 | 147 |
| 25–45 | 48.4 | 186 |
| 46–50 | 7.3 | 28 |
| 51 and Above | 5.7 | 22 |
| Missing | 0.3 | 1 |
| **Education** | | |
| Primary and Middle School | 1.8 | 7 |
| High School | 9.6 | 37 |
| Associate or Bachelor's Degree | 59.4 | 228 |
| Graduate Degree | 22.9 | 88 |
| Missing | 6.3 | 24 |
| **Occupation** | | |
| Government Officer | 20.8 | 80 |
| Private Sector | 3.9 | 15 |
| Retired | 2.3 | 9 |
| Small Business Owner | 2.9 | 11 |
| Student | 37.5 | 144 |
| Freelance | 6.3 | 24 |
| Manager | 5.5 | 21 |
| Other | 20.3 | 78 |
| Missing | 0.5 | 2 |
| **Income** | | |
| TRY 4400–4900 | 22.4 | 86 |
| TRY 4901–5500 | 19.3 | 74 |
| More than TRY 5500 | 45.1 | 173 |
| Missing | 13.3 | 51 |

According to the age data, the majority of respondents were between the ages of 25 and 45, with 38.3% being between the ages of 17 and 24, and respondents older than 46 whose age was more than 46 stood for 13% of total respondents. One respondent did not answer the age question, which was 0.3% of the total.

Continuously, the majority of the respondents were found to have an associate or bachelor's degree with 59.4%, followed by a graduate degree for 22.9%. The total percentage of primary, middle school, and high school graduates was found to be 11.4%. Lastly, 24 respondents did not answer the education question, which stood for 6.3%.

Furthermore, the majority of the respondents were students (37.5%), followed by government officers (20.8%). Additionally, 20.3% of the respondents stated that their occupation fell under "other". The remaining occupations comprised 20.9% of the total, and 2 people did not answer this question.

Lastly, the majority of the respondents declared that they received a monthly salary of more than TRY 5500.

Additionally, the respondents were asked about their TV watching habits as shown in Table 3, including how many hours a day they spent watching TV, which local TV channels they preferred, and the reason for watching the local TV channels they preferred.

**Table 3.** TV watching habits data.

| | | **Frequency** | **Percent** | **Valid Percent** | **Cumulative Percent** |
|---|---|---|---|---|---|
| | 1 Hour | 170 | 44.3 | 45.2 | 45.2 |
| | 2–3 Hours | 141 | 36.7 | 37.5 | 82.7 |
| | 4–5 Hours | 50 | 13.0 | 13.3 | 96.0 |
| Valid | 6 Hours | 7 | 1.8 | 1.9 | 97.9 |
| | More than 7 Hours | 8 | 2.1 | 2.1 | 100.0 |
| | Total | 376 | 97.9 | 100.0 | |
| Missing | System | 8 | 2.1 | | |
| Total | | 384 | 100.0 | | |

Firstly, according to the data about the daily TV viewing frequencies, the majority of the respondents stated that they watched TV for approximately one hour a day, and this was followed by 36.7% of the group of respondents stating that they watched TV 2–3 h a day. The total percentage of respondents who declared that they watched TV more than 4 h a day was 16.9%. Eight people did not want to answer this question.

Continuously, respondents were asked about the name of the local TV channels they mostly preferred to watch (as shown in Table 4). The results revealed that the majority (35.9%) of the respondents watched the BRTK channel, followed by 31.3% who stated that they did not prefer a specific TV channel.

Lastly, when the reason for "watching the local TV channel they prefer" was asked, the majority (52.9%) of the respondent stated that they preferred it for the "News", followed by 14.6% that stood for "no specific program preference". The other program preferences were sports, entertainment, and others, which summed up to 32.6% of the total as can be seen in Table 5.

### 3.6. The Validity and Reliability of the Measured Variables

Firstly, an exploratory factor analysis was conducted to reveal the factor structure of the items used in the measurement and to examine the questionnaire's internal reliability as can be seen from Table 6. When the exploratory factor analysis was carried out with a principal axis factoring extraction method and varimax rotation, it was possible to evaluate the accumulation of the variables under different components.

When the rotated component matrix was analyzed, the factors were found to mainly accumulate under three components, with factor loadings of 0.5 and above. However, four questions were found to have factor loadings that were less than 0.5, and they had to be removed, which included questions 6, 8, 9, and 10. Questions 1, 2, 3, 4, 5, 6, and 7

were grouped under one component; 11, 12, 13, 14, 15, and 16 were grouped under one component; and, lastly, questions 17, 18, and 19 were grouped under one component.

Table 7 below shows the remaining factors, with factor loading and groupings outlined as the three variables of the study.

**Table 4.** TV channel audience preferences.

| | | Frequency | Percent | Valid Percent | Cumulative Percent |
|---|---|---|---|---|---|
| | | **TV Channel** | | | |
| Valid | BRTK | 138 | 35.9 | 52.3 | 52.3 |
| | SİM | 14 | 3.6 | 5.3 | 57.6 |
| | KANAL T | 28 | 7.3 | 10.6 | 68.2 |
| | GENÇ TV | 43 | 11.2 | 16.3 | 84.5 |
| | KIBRIS TV | 16 | 4.2 | 6.1 | 90.5 |
| | DİYALOG | 9 | 2.3 | 3.4 | 93.9 |
| | ADA TV | 15 | 3.9 | 5.7 | 99.6 |
| | BRT 2 | 1 | 0.3 | 0.4 | 100.0 |
| | Total | 264 | 68.8 | 100.0 | |
| Missing | System | 120 | 31.3 | | |
| Total | | 384 | 100.0 | | |

**Table 5.** Audience program preference.

| | | Frequency | Percent | Valid Percent | Cumulative Percent |
|---|---|---|---|---|---|
| | | **Program Preference** | | | |
| Valid | News | 203 | 52.9 | 61.9 | 61.9 |
| | Sport | 27 | 7.0 | 8.2 | 70.1 |
| | Entertainment | 49 | 12.8 | 14.9 | 85.1 |
| | Other | 49 | 12.8 | 14.9 | 100.0 |
| | Total | 328 | 85.4 | 100.0 | |
| Missing | System | 56 | 14.6 | | |
| Total | | 384 | 100.0 | | |

**Table 6.** Factor analysis, rotated component matrix.

| | **Rotated Component Matrix** | | | |
|---|---|---|---|---|
| | **Factor Loadings** | | | |
| | **1** | **2** | **3** | |
| VAR1 | **0.538** | 0.205 | 0.203 | |
| VAR2 | **0.542** | 0.410 | 0.107 | |
| VAR3 | **0.598** | 0.120 | 0.128 | |
| VAR4 | **0.521** | 0.469 | 0.137 | |
| VAR5 | **0.504** | 0.461 | 0.167 | |
| VAR6 | 0.441 | 0.179 | 0.204 | Removed |
| VAR7 | **0.546** | 0.168 | 0.188 | |

**Table 6.** *Cont.*

| | Rotated Component Matrix | | | |
| --- | --- | --- | --- | --- |
| | Factor Loadings | | | |
| | **1** | **2** | **3** | |
| VAR8 | 0.358 | 0.450 | 0.145 | Removed |
| VAR9 | 0.414 | 0.285 | 0.177 | Removed |
| VAR10 | 0.268 | 0.495 | 0.281 | Removed |
| VAR11 | 0.176 | **0.754** | 0.113 | |
| VAR12 | 0.128 | **0.698** | 0.190 | |
| VAR13 | 0.153 | **0.602** | 0.121 | |
| VAR14 | 0.152 | **0.718** | 0.262 | |
| VAR15 | 0.234 | **0.606** | 0.206 | |
| VAR16 | 0.291 | **0.508** | 0.322 | |
| VAR17 | 0.201 | 0.174 | **0.803** | |
| VAR18 | 0.199 | 0.179 | **0.843** | |
| VAR19 | 0.197 | 0.219 | **0.807** | |
| | Extraction Method: Principal Axis Factoring | | | |
| | Rotation Method: Varimax with Kaiser Normalization | | | |

**Table 7.** Factor analysis remaining variable groups.

| Remaining Variable Groups | | | | | |
| --- | --- | --- | --- | --- | --- |
| Channel Image | | Program Quality | | Channel Preferance | |
| VAR1 | **0.538** | VAR11 | **0.754** | VAR17 | **0.803** |
| VAR2 | **0.542** | VAR12 | **0.698** | VAR18 | **0.843** |
| VAR3 | **0.598** | VAR13 | **0.602** | VAR19 | **0.807** |
| VAR4 | **0.521** | VAR14 | **0.718** | | |
| VAR5 | **0.504** | VAR15 | **0.606** | | |
| VAR7 | **0.546** | VAR16 | **0.508** | | |

Later, KMO and Bartlett's test was performed for the remaining 15 questions in order to reveal how suited the data were and whether the variances were equal for all samples as shown in Table 8.

**Table 8.** Factor analysis and KMO and Bartlett's test.

| Variable | Kaiser–Meyer–Olkin Measure of Sampling Adequacy | Bartlett's Test of Sphericity | | |
| --- | --- | --- | --- | --- |
| | | Approx. Chi-Square | df | Sig. |
| **Channel Image** | 0.845 | 708.459 | 15 | 0.000 |
| **Program Quality** | 0.792 | 831.002 | 15 | 0.000 |
| **Channel Preference** | 0.761 | 862.007 | 3 | 0.000 |

For the channel image variable, the Kaiser–Meyer–Olkin measure of sampling adequacy was found to be satisfying (0.845), and the significance was 0.000.

The KMO and Bartlett test result for the questions for the program quality variable came out to be satisfying, whereby, sampling adequacy was 0.792 and significance was 0.000.

Lastly, the KMO and Bartlett result for the consumer preference variable was found to also be sufficient (sampling adequacy 0.761 and Sig. 0.000).

Continuously, in order to confirm the convergent validity of the constructs, the composite reliability (CR) and average variance extracted (AVE) values were calculated. Additionally, a reliability analysis was performed and Cronbach's alpha values were calculated for the 15 questions.

All the composite reliability results for the three variables were found to be good (higher than 0.6). The average AVE values for channel image and program quality were found to be less than 0.5; however, because AVE values were higher than 0.6, the convergent validity of the constructs was acceptable as can be seen from Table 9.

**Table 9.** Composite reliability, average variance extracted, and Cronbach's alpha values.

| Variables | Content (Statements about a Specific TV Channel by Respondents) | CR | AVE | $\alpha$ |
|---|---|---|---|---|
| Channel Image | VAR 1: It appeals to all segments of the society (rich, low-income, etc.) | 0.713 | 0.352 | 0.820 |
| | VAR 2: It is a lively and entertaining channel | | | |
| | VAR 3: It acts with social responsibility awareness and is sensitive to social problems | | | |
| | VAR 4: It is a creative, innovative, pioneering channel | | | |
| | VAR 5: There is a wide variety of programs and there are programs for all age groups | | | |
| | VAR 7: It is a reliable institution | | | |
| Program Quality | VAR 11: Domestic serial shows are popular and interesting productions | 0.814 | 0.426 | 0.819 |
| | VAR 12: Magazine and entertainment programs are of high quality | | | |
| | VAR 13: It has quality domestic and foreign movie broadcasts | | | |
| | VAR 14: News and news programs reveal the truth | | | |
| | VAR 15: Its news and news programs are neutral and objective | | | |
| | VAR 16: It has good quality and informative sports programs | | | |
| Channel Preference | VAR 17: Compared to others, it is the channel I watch most regularly | 0.858 | 0.668 | 0.922 |
| | VAR 18: It is the channel I watch most frequently when compared to others | | | |
| | VAR 19: It is my favorite channel when compared to others | | | |

## 4. Results

### 4.1. Results of Correlation Analysis

The Pearson correlation analysis, as shown in Table 10 below, was performed to evaluate the correlations between the channel image, program quality, channel preference, and the moderator variable, which was the TV viewing frequency. The correlations between all the variables were found to be significant at the 0.01 level, except for the correlation between the TV viewing frequency and channel image, which was found to be significant at the 0.05 level.

Firstly, the Pearson correlation between channel image and program quality showed that there was a significant (0.01 level), positive, and strong correlation between the two variables (0.733). According to this result, it was possible to accept the first hypothesis of the study "H1: There is a relationship between channel image and program quality".

Secondly, to understand whether the second hypothesis was supported or not, the correlation level between the program quality and channel preference had to be reviewed. According to the Pearson correlation, there was a significant (0.01 level), positive, and moderate correlation between the two variables (0.580). This result revealed that the second hypothesis, "H2: There is a relationship between program quality and channel preference", was supported.

**Table 10.** Descriptive statistics and correlations analysis.

| | Descriptive Statistics | | Correlations | | | | |
|---|---|---|---|---|---|---|---|
| | Mean | Std. Dev. | Channel Image | Program Quality | Channel Preference | Mod_Varf | Frequency |
| Channel Image | 3.218 | 0.799 | 1 | | | | |
| Program Quality | 3.071 | 0.818 | 0.733 ** | 1 | | | |
| Channel Preference | 3.464 | 1.114 | 0.543 ** | 0.580 ** | 1 | | |
| MOD_VARF | 5.705 | 3.788 | 0.483 ** | 0.427 ** | 0.289 ** | 1 | |
| Frequency | 1.807 | 0.905 | 0.116 * | 0.142 ** | 0.097 | 0.811 ** | 1 |

*, Correlation was significant at the 0.05 level (2-tailed); **, Correlation was significant at the 0.01 level (2-tailed).

Lastly, the correlation between the channel image and channel preference was analyzed. The result showed that there was a significant (0.01 level), positive, and moderate correlation between the two variables (0.543). By this result, it was possible to conclude that the third hypothesis, "H3: There is a relationship between channel image and channel preference", was accepted.

### 4.2. Fit of Entire Model

Based on the theoretical model, a research model was established on IBM-AMOS (Figure 2). Here, "Channel Image", "Program Quality", and "Channel Preference" were all latent variables with subcomponents. "Daily Viewing Frequency" was a directly measured variable, and, hence, was represented in a square. As proposed in AMOS [36], an additional interaction term was added labeled "MODVARf". It was calculated by multiplying the "Channel Image" and "Daily Viewing Frequency" and placing the result in the model. This model was tested for model fit indices, for which the maximum likelihood (ML) parameter estimation method was used. With all values exceeding the cut-off values, the adequacy of the research model was established. According to Table 11 below, the model had a good fit.

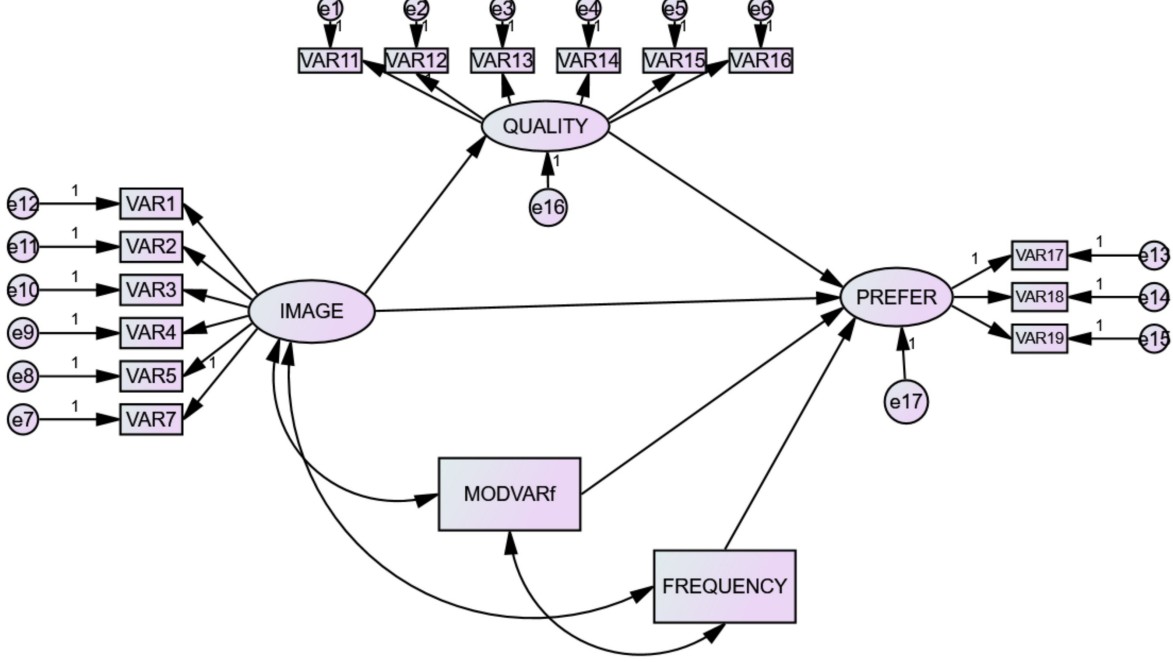

**Figure 2.** The structural model for hypotheses testing.

Model fit indicators are shown in Table 11 below:

**Table 11.** Fit of entire model.

|  | CMIN/DF | GFI | AGFI | CFI | RMSEA | IFI |
|---|---|---|---|---|---|---|
| **Model 1** | 2.120 | 0.934 | 0.903 | 0.967 | 0.054 | 0.967 |
| **Threshold** | <5 | >0.85 | >0.9 | >0.9 | <0.08 | >0.9 |

Note: good fit.

*4.3. Results of Structural Modeling Analysis*

After ensuring a good model fit, the established model was run in AMOS, and standardized estimates were calculated, as depicted in Figure 3.

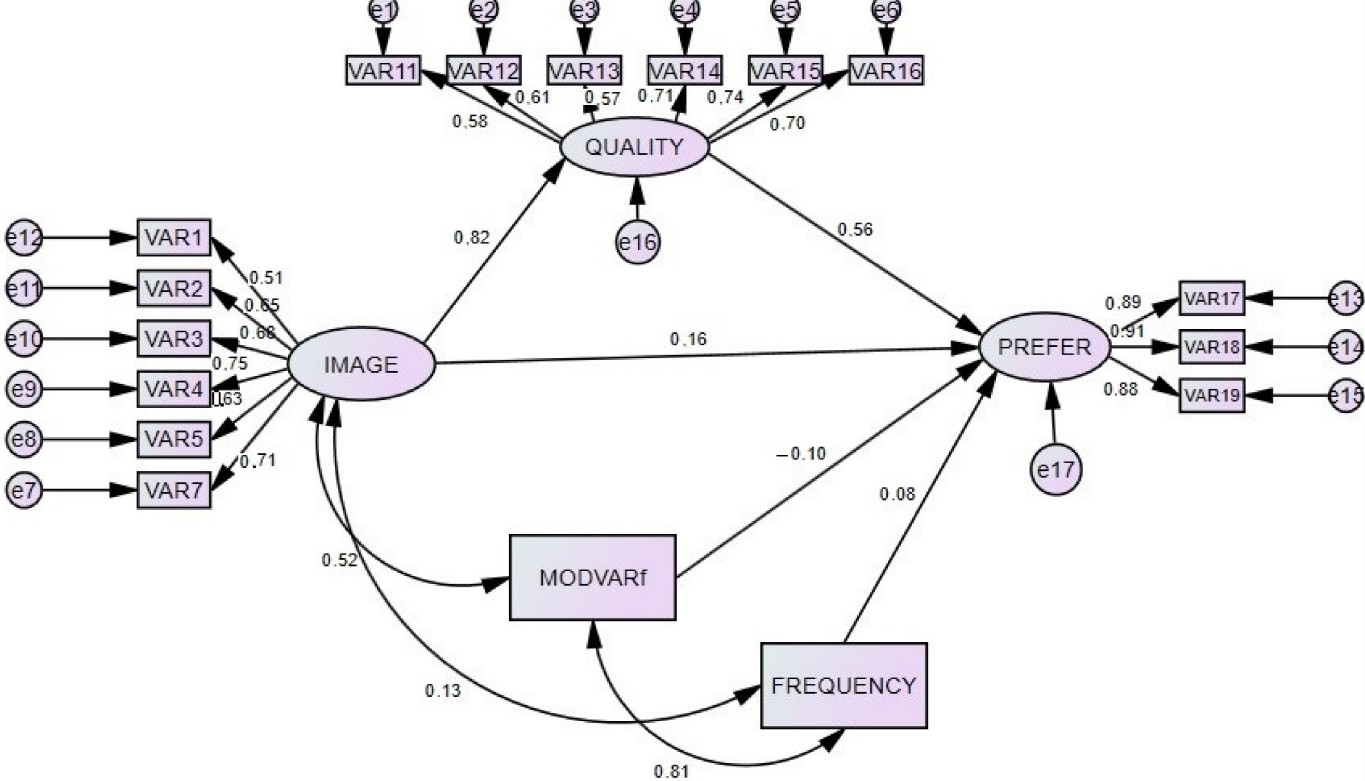

**Figure 3.** The structural model for standardized estimates.

Among all the hypotheses formulated in this study, the hypothesis test results for those regarding direct effects were as follows: (1) H1—1: the effect of the channel image on the perceived program quality was supported statistically (estimate 0.818 and *p* value less than 0.01); (2) H1—2: the effect of the channel image on the channel preference was rejected statistically (estimate 0.162 and *p* = 0.206 > 0.05); (3) H1—3: the effect of the program quality on the channel preference was accepted statistically (estimate 0.558, *p* < 0.001); (4) H1—4: the moderation effect of the viewing frequency on the relationship between the channel image and the channel preference was rejected (estimate −0.097 and *p* = 0.396 > 0.05).

The results showed that the channel image had no direct effect on the channel preference; however, the relationship between these two variables was mediated by the program quality. Additionally, the hypothesized moderation effect of the TV viewing frequency was rejected statistically.

Estimates (Table 12).

**Table 12.** Results of analysis on direct effects.

| Standardized Regression Weights: (Group Number 1—Default Model) | | | | | |
|---|---|---|---|---|---|
| | | | Estimates | *p* | |
| QUALITY | <— | IMAGE | 0.818 | *** | Supported |
| PREFR | <— | IMAGE | 0.162 | 0.206 | Not supported |
| PREFR | <— | QUALITY | 0.558 | *** | Supported |
| PREFR | <— | MOD_VARIABLE | −0.097 | 0.396 | Not supported |
| PREFR | <— | FREQUENCY | 0.082 | 0.391 | Not supported |

## 5. Conclusions and Practical Implications

Thus far, there was no empirical study in this field regarding the relationships between a TV channel image, program quality, viewing frequencies, and channel preference of audiences. There is only one study by [4] with similar variables, and there are very few studies about the viewing rates of channels; more specifically, about the program rating scores. The present study tried to understand the relationship between channel image and channel preference, which is an important factor for the sustainability of TV channels in the sector, while considering the hypothesized mediation effect of program quality and the moderation effect of viewing frequencies on this relationship. Channel preference, or, in other words, the preferences/choices of viewers about TV channels, is an important concept for the TV sector, especially in North Cyprus, where the TV sector is small and there is significant local competition. TV channels do not target a specific audience and all their programs are for a general audience. These local TV channels strive to achieve the preference of the audience so they can remain in the competition and sustain themselves in the sector without having any significant differences from each other. Therefore, it is important to understand what kind of factors are related to the preference of the audience and, mainly, which of these factors affect their choices. The idea behind this study was, mainly, to understand how the channel image affected the preferences of the audience, while looking at the relationships among the channel image, program quality, and channel preference overall, and, then, to see if there was a mediation effect of program quality and a moderation effect of viewing frequencies on this relationship. From the marketing perspective, it is possible to consider each TV channel company as a brand with an image in the minds of the audience, with each TV audience having the potential to be accepted as a customer. Furthermore, the programs produced and broadcast by TV channels should be considered as products of these brands, and the choices of the audience about the programs can be considered as the product choice [4]. Understanding these relationships is very important for the managers and owners of the TV channels to shape their investment decisions to improve their channel image, engage more audience activity, keep their market share, and sustain in the market, especially in small markets such as North Cyprus. TV channels need to determine their marketing strategies according to the tastes of the audience.

Firstly, according to the results of the correlation analysis, it was concluded that there was a positive and strong correlation between the channel image and program quality. This result revealed that if the viewers observed the channel image better, they tended to perceive the program quality better as well, and vice versa; if the viewers believe the TV channel had good-quality programs, they would have a better perception of the TV channel's image in their minds. In studies from the extant literature, the effect of brand image on perceived quality was also investigated, and it was concluded that brand image positively affected a perceived product or service quality [9,10], which was in line with the findings of the present study. If the TV channel could create a positive image in the audience's minds, the perceived program quality of the audience about the programs of this channel would also be high [4].

Continually, it was also revealed that there was a positive and moderate correlation between the program quality and channel preference, which could be interpreted as that if the audience had a positive view about the program quality of the TV channel, they tended to choose to watch that TV channel more, and if they preferred to choose a TV channel, it means that they had a positive observation about the program quality of the TV channel. From the perspective of TV channels, if we considered that sustainable consumer behavior required a long-term commitment in behavioral and attitudinal terms, in this sense, a repeating and long-term channel preference from the audience would increase the possibility of the development of brand loyalty [4]. Additionally, brand image is expected to influence the loyalty of consumers [37]. In other words, the better the image the audience has of the TV channel and its programs, the more they tended to choose to watch more, and, as they watched more, they tended to develop a loyalty to the TV channel, resulting in the TV channel views increasing and them becoming more competitive in the sector against the other channels.

Lastly, it was found that there was a positive and moderate correlation between the channel image and the channel preference of the audience. In other words, as stated in the previous studies, the brand image of a company has a positive relationship with consumer preference for the product [38]. This result could be interpreted as that if the viewers had a positive TV channel image, they tended to prefer that TV channel more, and, on the other hand, if viewers preferred a TV channel more, this meant that they had a good perception of the TV channel's image. The results could be supported by the idea that " an increase in all positive subjective and perceptual associations created in the mind of the consumers would increase the consumer's brand preference". The preference for a channel with a better brand image would increase, and if the image was negative, the preference of the audience for this channel could also be negatively affected [4]. Additionally, studies in the extant literature supported the hypothesis that brand image has a positive and significant influence on brand preference [25,39,40].

Furthermore, according to the results from the SEM analysis, the effect of the channel image on the perceived program quality was supported statistically, and the effect of the program quality on the channel preference was also accepted statistically. However, the direct effect of the channel image on the channel preference was rejected. The results revealed that even though the channel image had no direct effect on the channel preference, the program quality was a result of the channel image, and the channel preference was a result of the program quality. This result was in line with the studies in the literature in terms of how the relationship between perceptions of consumers and their purchase intentions was mediated by perceived quality [41]. (1) When the results were evaluated, the effect of the channel image on the perceived program quality was supported with an estimate value of 0.818, and the effect of the channel image on the channel preference was rejected statistically because $p = 0.206 > 0.05$, with an estimated value of 0.162. Lastly, the effect of the program quality on the channel preference was accepted statistically, with an estimated value of 0.558. Even though there was no previous study regarding the mediation effect of the program quality on the relationship between the channel image and channel preference, some results from the extant literature concluded that the quality of a program's broadcast on TV channels may affect the image of the TV channel, and since the image of a TV channel also affects the perceived program quality, it can be argued that there is a reciprocal causality between the brand image and perceived product quality for TV channels [4], as found in the current study, whereas the program quality was a result of the channel image.

Lastly, the results were analyzed to understand whether audience viewing frequencies had a moderation effect on the relationship between the channel image and channel preference. According to the results, the moderation effect of the viewing frequencies on the relationship between the channel image and channel preference was rejected with $p = 0.396 > 0.05$ and an estimated value of 0.097. From the previous findings of the present study, it was possible to conclude that the channel image was a significant predictor of

the program quality, and the program quality was a predictor of the channel preference; however, the results of the statistical analysis revealed that the viewing frequencies did not act like a moderator on the relationship between the channel image and channel preference. In other words, the viewing frequency of the audience did not have an interaction effect on the relationship, so it did not change the nature of the relationship between the predictor and the outcome variable.

*Practical Implications*

These findings indicated profound practical results for decision makers in media organizations.

Overall, according to the results of the study, it was possible to conclude that the channel image affected the TV channel preferences of the audience, not directly, but through the program quality, which is important for TV channel companies to consider when determining their marketing strategies. Instead of pouring scarce resources into efforts to raise the viewing frequency, they must focus on increasing the "Program Quality", and the advertising themes must focus on emphasizing the "Program Quality" instead of simply repeating slogans. The brand image of the TV channel is expected to affect how the audience perceives the program quality of the channel [9]. Additionally, according to the extant literature, the program quality is very crucial; in other words, one of the most important ways of achieving public interest through TV broadcasting is to ensure that the programs broadcast are of high quality [42]. Considering that the preference of the audience plays a significant role in the success of TV companies in the sector, in order to have an advantage over their competitors, it is important for the managers and owners to understand what factors affect the viewers' preferences, so that they can shape their strategies and investment plans accordingly. Regarding the findings of the current study, the managers of TV channels need to take into account that a positive channel image is a predictor of an increased audience preference, along with good program quality, which can influence the channel image's effect on preference.

## 6. Limitations and Future Research

The main limitation of the study was the sample size, which represented the audience of North Cyprus viewers of local TV channels. The data were collected through a convenience sampling method, which allowed for the researcher to reach respondents easier; therefore, it was susceptible to selection bias. However, the researcher tried to equally reach each respondent category, such as concerning the age, gender, and occupation. In addition to this, another limitation of the study was that the control variables, including gender, age, and monthly income, which were not included in the analysis. Lastly, the data were collected with the help of a questionnaire only, which was a limited way of elaborating in-depth information from the respondents. Altogether, the findings of the current study cannot be generalized to broader populations, meaning that it is useful only for the current universe of the study.

In future research, a method of probability sampling could be used to select the respondents to eliminate any possible bias that comes with nonprobability sampling methods. Additionally, data could be collected from respondents using interviews to collect more in-depth information. In addition, future research can utilize control variables to gain more insight into the differences in audiences, and how those differences affect their channel preference.

**Author Contributions:** Methodology, T.A. and A.B.; Software, T.A.; Formal analysis, T.A.; Investigation, A.D.; Resources, A.D.; Writing—original draft, A.D.; Writing—review & editing, A.B.; Supervision, T.A. and A.B.; Project administration, A.D. All authors have read and agreed to the published version of the manuscript.

**Funding:** This research received no external funding.

**Institutional Review Board Statement:** Not applicable.

**Informed Consent Statement:** Informed consent was obtained from all subjects involved in the study.

**Data Availability Statement:** Data can be obtained from corresponding authors upon request.

**Conflicts of Interest:** The authors declare no conflict of interest.

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
