# Peer review of "Factors Affecting the Sustained Market Position of TV Channels: Insights from Audience Preferences"

_sustainability, doi:10.3390/su142316138_

Round 1

Reviewer 1 Report

Thank you for the article. The article is readable, but I find some passages very overlapping and the same arguments are repeated over and over again. I would recommend defining the variables better. What exactly is meant by quality, image, whether the respondents understand these concepts in the same way as the author. The conclusions of the article are described by the author as significant, but I think that every TV owner is trying to be as good as possible in terms of image and production quality, so I don't see much impact for practice here, because it is automatic that everyone is trying to be as good as possible and it is logical that a good image and production quality helps significantly. I appreciate the statistical treatment of the data.

Author Response

Thank you for all the comments. All the answers can be found in the attached file.

Reviewer 2 Report

The paper stands in the perspective of viewers' preferences and tries to explore the factors affecting the sustained market position of TV channels through empirical research. It has some theoretical and practical implications, but the following problems exist:

1. In the introduction section, the first paragraph is too long, and the authors are advised to adjust it; and the introduction occupies a large amount of space in the paper compared to the other sections, making the core issues and contributions of this research not prominent. It is suggested that the authors make adjustments.

2. In the Theoretical background and hypothesis development section, the authors are unclear about the proposed hypothesis, and it is suggested that the authors express the relationship between the influencing factors specifically.

3. In the Research Methods section, the authors are advised to correlate the research methods with the hypotheses presented in this paper.

4. It is suggested that the authors add a robustness check to the article to confirm the reasonableness of the hypotheses and the accuracy of the findings of the study.

5. In the Conclusion and Practical Implications section, it is suggested that the authors highlight the findings of this paper.

6. There are some typographical errors and punctuation mistakes in this paper, it is recommended that the author check and correct them carefully.

Based on the above comments, and given the academic standard of this journal, it is recommended a revision.

Author Response

Thank you for all the comments. The answers can be found in the attached file.

Reviewer 3 Report

This paper studies the factors Affecting the Sustained Market Position of TV Channels giving Insights from Audience Preferences. It is a well written work which has a good, organized structure. Also, the topic can be very interesting to most readers, since not much research has been conducted so far examining the combination of these certain factors.  The authors highlight some interesting points that pose value and the literature review is well promised. In its present form the paper, in my opinion, it could be suitable for publishing after the revision of the following issues:

1. The section “Sustainability in the TV sector” needs to be strengthened with relevant research about the topic. The way it is written, without more references, looks like author assumptions.  Also, it would be useful to present relevant cases of small TV markets in a EU level.

2. The authors compare the small TV market of the island with the Turkish market but the characteristics of the TV market at North Cyprus are poorly discussed. How many free TV channels exist? Which is their profile/content related to channel image? How many pay TV channels are there? Which are the characteristics of the local audience?

3. In the research gap section [line 134] they mention “one study which was carried out in Turkey, looking at the relationship between TV channels image, program quality and channel preference”. Which is this study? The reference should be written.

I hope these comments will help the authors enhance the strength of their work.

Author Response

(The authors gave the same response as above.)

Round 2

Reviewer 2 Report

The authors have revised the paper well, according to my suggestiones.